# Transportation of Single-Domain Antibodies through the Blood–Brain Barrier

**DOI:** 10.3390/biom11081131

**Published:** 2021-07-31

**Authors:** Eduardo Ruiz-López, Alberto J. Schuhmacher

**Affiliations:** 1Molecular Oncology Group, Instituto de Investigación Sanitaria Aragón (IIS Aragón), 50009 Zaragoza, Spain; eruiz@iisaragon.es; 2Fundación Aragonesa para la Investigación y el Desarrollo (ARAID), 500018 Zaragoza, Spain

**Keywords:** single-domain antibodies, nanobody, VNAR, blood–brain barrier, transcytosis, nanoparticles, cell-penetrating peptides, carriers

## Abstract

Single-domain antibodies derive from the heavy-chain-only antibodies of *Camelidae* (camel, dromedary, llama, alpaca, vicuñas, and guananos; i.e., nanobodies) and cartilaginous fishes (i.e., VNARs). Their small size, antigen specificity, plasticity, and potential to recognize unique conformational epitopes represent a diagnostic and therapeutic opportunity for many central nervous system (CNS) pathologies. However, the blood–brain barrier (BBB) poses a challenge for their delivery into the brain parenchyma. Nevertheless, numerous neurological diseases and brain pathologies, including cancer, result in BBB leakiness favoring single-domain antibodies uptake into the CNS. Some single-domain antibodies have been reported to naturally cross the BBB. In addition, different strategies and methods to deliver both nanobodies and VNARs into the brain parenchyma can be exploited when the BBB is intact. These include device-based and physicochemical disruption of the BBB, receptor and adsorptive-mediated transcytosis, somatic gene transfer, and the use of carriers/shuttles such as cell-penetrating peptides, liposomes, extracellular vesicles, and nanoparticles. Approaches based on single-domain antibodies are reaching the clinic for other diseases. Several tailoring methods can be followed to favor the transport of nanobodies and VNARs to the CNS, avoiding the limitations imposed by the BBB to fulfill their therapeutic, diagnostic, and theragnostic promises for the benefit of patients suffering from CNS pathologies.

## 1. Introduction

Single-domain antibodies constitute an attractive alternative to conventional monoclonal antibodies. In the past years, their research and applications in the biomedical field are growing exponentially. Among single-domain antibodies, nanobodies present structural and molecular properties resulting in unique properties such as high affinity, specificity, and biodistribution. Importantly, nanobodies can be engineered and tailored into wide-ranging formats to broaden their uses [1,2,3].

While some nanobodies are described to cross the blood–brain barrier (BBB), it still represents a challenge to fulfill their in vivo diagnostic and therapeutic potential in the treatment central nervous system (CNS) diseases [4,5,6].

While several recent reviews have explored the different ways single-domain antibodies reach the brain [4,5,6], mainly through transcytosis, the present report reviews the up-to-date status of current literature on the transport of nanobodies through the BBB, providing an additional overview of coadjutant methods to deliver nanobodies to the brain parenchyma and discussing several strategies that could be applied to direct nanobodies towards the BBB, as well as some of the potential challenges to be addressed for clinical translation.

## 2. The Blood–Brain Barrier

The blood–brain barrier (BBB) is a highly selective and regulated filter that controls and restricts the passage of substances between blood circulation and brain parenchyma. The BBB is an extensive blood capillary network with a total surface area of 20 m^2^ [7]. The BBB comprises multiple cell types closely related to each other within the so-called neurovascular units (Figure 1). These include brain microvascular endothelial cells (BMECs), pericytes, astrocytes, microglia, neurons, and extracellular matrix components [8,9].

The intimate association of the neurovascular unit cells is recognized as the functional structure for regulating cerebral blood flow. Among those cell types, BMECs form adherent and tight junctions forcing transcytosis as the unique mechanism for the transport of molecules from the bloodstream through the capillary wall into the central nervous system (CNS) or vice versa, except for those small and lipophilic [10]. Moreover, BMECs express multiple broad-spectrum efflux pumps—including P-glycoprotein and alpha(2,3)-sialoglycoprotein receptor—actively preventing the passage of many molecules via passive diffusion through the BBB, including metabolic waste products. It is estimated that the BBB blocks the transport of ∼98% of molecules, thus representing a significant challenge for treating brain diseases [11].

The integrity of anatomical barriers is a hallmark of health [12]. The BBB protects the CNS from infections and toxic substances such as endogenous metabolites, xenobiotics, or exogenous neurotoxic substances in the bloodstream [13]. Lack of integrity and dysfunction of the BBB are associated with numerous neurological diseases and brain pathologies, including cancer, thus resulting in BBB leakiness [14,15,16,17]. This could partially explain why several drugs with limited BBB permeability have nevertheless shown clinical efficacy in some situations. For example, the immune-checkpoint inhibitors Nivolumab and Ipilimumab, anti- Programmed Death 1 (PD-1) and anti- Cytotoxic T-Lymphocyte Antigen 4 (CTLA-4) monoclonal antibodies, have shown clinical benefit akin to their efficacy for other sites of non-CNS metastasis in patients, including some with small, asymptomatic melanoma-to-brain metastases [18,19,20].

Aberrant endothelial-pericyte and/or astrocyte-pericyte signaling can result in BBB dysfunctions, causing the local accumulation of blood-derived neurotoxins and the reduced extrude of neurodegeneration-associated proteins in a complex self-amplificatory system influenced by genetic and environmental risk factors, arterial hypertension, and lifestyle [17,21]. For example, the E4 variant of apolipoprotein E (ApoE4), considered the main susceptibility gene for Alzheimer’s disease, leads to an accelerated breakdown of the BBB and degeneration of brain capillary pericytes [21,22].

While the BBB restricts the entry of many cells and most macromolecules, it is not an impenetrable barrier to the transmigration of metastasizing cancer cells [19,23]. Indeed, the restrictive features of the BBB play a protective role for the metastatic cells that effectively breach the BBB. In this regard, the brain can function as a sanctuary site for these metastatic cells, remaining protected against immune cells, chemotherapy, and other toxic agents once metastatic cells have reached the cerebral compartment [24,25].

## 3. Antibodies, Nanobodies and the BBB

Papain enzymatic digestion of a classical immunoglobulin IgG produces three fragments of similar MW (50 kDa). Two of them, termed fragments of antigen-binding (Fab), retain the antigen-binding affinity. The third one, named fragment crystallizable region (Fc region), does not bind the antigen and mediates effector functions for humoral immune response [26,27]. The Fc region interacts with cell surface receptors (Fc receptors) and complement proteins.

In homeostasis, the integrity of the BBB may difficult the passage of conventional antibodies to spontaneously cross the BBB due to the Fc-receptor-mediated efflux to the blood (Figure 2). Permeability of the BBB is limited to receptor-specific ligands or molecules, which show both lipophilic solubility and a MW smaller than 400 Da, hindering therapeutic concentrations in the brain [28]. Under these restrictive conditions, only 0.01–0.4% of proteins in the blood can enter the CNS through passive diffusion, including some therapeutic antibodies which often are IgG isotype (150 kDa) [29]. Despite the controversy, some authors claim that the binding of the Fc domain to Fc receptors in the BBB could represent a potential route for brain access and/or exit. It has been proposed that an Fc receptor triggers the reverse transcytosis of IgG across the BBB, from the brain to blood [30]. In addition, the presence of Fcγ receptor (FcγR) in the CNS could be a possible contributor to the receptor-mediated immune clearance of monomeric or oligomeric IgG complexes [31]. Furthermore, the neonatal Fc receptor (FcRn) located at the brain microvascular endothelium actively mediates the reverse transcytosis of IgG from the brain to blood across the BBB [32,33]. Nonetheless, recent studies have suggested that neither FcγR nor FcRn significantly contribute to the brain clearance of IgG. IgG transcytosis occurs by non-saturable, non-specific mechanisms suggesting fluid-phase endocytosis like macropinocytosis [34,35,36].

Due to the limited permeability of drugs at the BBB, many targeted therapies fail to treat CNS diseases. To overcome this problem, novel therapy approaches for the treatment and diagnosis of CNS diseases are being investigated. Nanobodies and variable new antigen receptor (VNARs), the antigen-binding fragments of heavy-chain-only antibodies, emerge as promising alternatives for theragnosis. Nanobodies are the single variable domain (VHH) of the heavy-chain-only antibodies (HCAbs) of *Camelidae* (camel, dromedary, llama, alpaca, vicuñas and guananos) [39], while VNARs are the single variable domain of the heavy-chain-only antibodies or immunoglobulin new antigen receptors (IgNARs) of cartilaginous fish [40] (Figure 2). Both single-domain antibody fragments constitute the smallest intact functional antigen-binding recombinantly-isolated molecules (diameter of 2.5 nm and height of 4 nm; 12–15 kDa) [41]. Nanobodies present three complementarity determining regions (CDRs) harbored by four conserved sequence framework regions (FR1-4). Compared with human antibodies, nanobodies contain more extended CDR1 and CDR3, being the CDR3 the main contributor to antigen-binding [41,42]. VNARs present a similar extended CDR3 region with additional disulfide bridges while CDR2 is absent [43].

Small size and increased plasticity provide nanobodies with an outstanding potential to recognize unique conformational epitopes, such as unstructured regions of intrinsically disordered proteins [44], and active sites of enzymes and cavities of receptors [45,46]. Nanobodies have an outstanding antigen specificity with binding affinities in the nanomolar or even picomolar range, with rapid renal clearance (t_1/2_ = 30–60 min) from the blood [1,2,47]. Nanobodies show high solubility and chemical and thermal resistance. Furthermore, clinical applications are possible due to their limited immunogenicity, feasibility and easy and cost-effective production using recombinant expression [1,2,3,48]. Humanization of nanobodies by introducing genetic mutations in their structure have been employed to minimize immunoresponse towards their application in clinics [49,50]. Recently, ALX-0681 (Caplacizumab) the first nanobody reached the clinic for the treatment of thrombotic thrombocytopenic purpura patients, representing a milestone in the nanobodies’ field [51,52].

However, their small size and increased frequency of polar and charged amino acids could represent limitations for the use of nanobodies for brain diagnostic and therapeutic applications. Their fast renal clearance from blood restricts their plasma concentration, and therefore therapeutic levels in the brain. Strategies to improve nanobody pharmacokinetics include covalent attachment of poly (ethylene glycol) (PEG) [53], fusion with Fc domains [54,55,56], or generation of multivalent or multiparatopic constructs, including fusions with nanobodies targeting serum albumin [57,58,59], which also enabled the coupling of additional binding arms increasing affinities to targets. The penetration rate of nanobodies through the BBB is also limited by their surface charge, rendering difficulties accessing the brain when negative [60,61]. The concentration of nanobodies in the CNS is reduced by half, 24 h post-injection [62].

## 4. Brain Delivery of Nanobodies through the Blood–Brain Barrier

Several reviews have described many aspects of the BBB drug delivery mechanisms employed to travel across, and detailed some routes that nanobodies use to traverse the BBB [4,5,6,63,64]. Here we present an up-to-date systematic review to cover the different strategies and methods to deliver nanobodies into the brain parenchyma.

### 4.1. Physiopathological Disruption of the BBB

Nanobodies and VNARs can reach the brain parenchyma when the BBB integrity is compromised as seen in numerous pathological conditions including cancer, inflammation, and infections [14,15,16,17]. It is well described that some brain tumors (both primary and secondary) can impair the BBB integrity becoming leaky for some molecules. Several nanobodies have been employed as molecular imaging probes to detect brain tumor lesions taking advantage of the local disruption of the BBB caused by high-grade gliomas and/or brain metastases. A human epidermal growth factor receptor 2 (HER2)-targeting nanobody (radiopharmaceutical ^68^Ga-NOTA-Anti-HER2 VHH1) that has proved efficacy to diagnose primary breast carcinoma by Positron Emission Tomography (PET)/Computed Tomography (CT) imaging in a phase I study [65] is being evaluated to detect breast to brain metastasis in phase II clinical trials (ClinicalTrials.gov NCT03331601). Several examples are found in preclinical settings. Monovalent (EG2), bivalent (EG2-hFc), and pentavalent (V2C-EG2) nanobodies directed against the epidermal growth factor receptor (EGFR) and its variant III (EGFRvIII) have been successfully employed for in vivo fluorescence imaging of orthotopic high-grade glioma-bearing mice [57]. Similarly, a nanobody directed against insulin-like growth factor-binding protein 7 (IGFBP7; VHH 4.43), was able to selectively detect glioblastoma (GBM) blood vessels after systemic injection in orthotopic GBM mouse models [66].

Some inflammatory processes can compromise the BBB integrity. Systemic inflammation, including systemic infections, can result in the alteration of the BBB functionality. The physiological permeability status of the BBB is altered by the underlying disruptive and non-disruptive changes of inflammation events [67]. VCAMelid or cAbVCAM1–5, a nanobody targeting mouse vascular cell adhesion protein 1 (VCAM-1), and the bivalent (BiVCAMelid) or bispecific (VCAM/ALB8) derivates achieved maximum CNS concentrations in focal brain vascular inflammation-induced models [59]. Other nanobodies have been found in the brain upon systemic administration. For example, the TNF Receptor-One Silencer (TROS), a trivalent nanobody developed by linking two anti-human tumor necrosis factor receptor (TNFR)-1 nanobodies with an anti-albumin arm (Nb Alb-70-96), has been investigated for the selective inhibition of this pro-inflammatory pathway in autoimmune disease of the CNS in preclinical settings in rodents [68]. Of importance, in a multiple sclerosis mouse model with experimental autoimmune encephalomyelitis (hTNFR1 Tg mice), the immuno-PET radiotracer ^99m^Tc-TROS had a significantly higher brain uptake and presence in the CSF than healthy mice, indicating that TROS enters the CNS not only, but mainly, when the BBB is altered [69]. The ability of TROS to inhibit TNF/TNFR1 signaling has also been studied in an Alzheimer’s disease mouse model. However, TROS was administrated intracerebroventricularly together with Aβ1–42 oligomers [70]. Other nanobodies have been generated but require BBB permeability studies in vivo. For example, V31-1 nanobody recognized intraneuronal Aβ42 oligomers, inhibited the formation of Aβ42 fibrils, and had therapeutic effects in vitro [71]. V31-1 has been used to detect the presence of Aβ42 oligomers in immunofluorescence assays of tissues from Alzheimer’s disease mouse models, although in vivo studies are still pending [72].

Several nanobodies can cross the BBB and target brain infections. A variety of nanobodies neutralizing *Lyssavirus* (Rabies virus, RSV) were isolated from a llama phage display library obtained after active immunization with RSV FTM-protein or Inactivated Rabies Vaccine Mérieux HDCV (Rab-C12, Rab-E6, Rab-E8, Rab-F8, and Rab-H7). Bivalent and biparatopic constructs had a significantly higher targeting capacity of the trimeric envelope protein (G protein) of *Lyssavirus*, showing the biparatopic combination Rab-E8/H7 and Rab-E6/H7 potent synergistic effects [58]. Systemic administration of Rab-E8/H7, including a trispecific nanobody containing an anti-albumin arm (Rab-E8/H7-ALB), revealed rapid influx into the brain in control mice [73]. Pentavalent nanobodies targeting the same Rabies virus antigen (combodies 26424 and 26434), stabilized by the fusion with a coiled-coil peptide derived from COMP48, demonstrated partial protective effects upon co-administration with the virus in the hind leg of mice [74]. Even though multivalent constructs showed enhanced efficacy and potency, mechanisms underlying their BBB penetrance are still unknown, and pharmacokinetics similar to antibodies have been proposed. Prolonged plasma half-life due to their higher MW may explain, in part, a higher brain exposure rather than better brain uptake.

The brain uptake of some nanobodies can be increased when the BBB it is altered due to infections. For example, Nb_An33, a therapeutic nanobody against the AnTat1.1 variant-specific surface glycoprotein (VSG) from *Trypanosoma brucei*, increased BBB penetration during the encephalitic stage of trypanosomiasis in rats [75]. 

### 4.2. Device-Based and Physicochemical Disruption of the BBB

Several strategies have been developed to favor the concentration of therapeutic agents into the CNS. Convection-enhanced delivery (CED) is a direct method of drug delivery to the CNS through one to several intraparenchymal microcatheters placed stereotactically into the brain parenchyma [76]. These microcatheters are connected to mechanical pumps to provide a continuous, positive-pressure micro-infusion of desired agents through target tissues allowing directed distribution along with large brain volumes and minimizing systemic side effects. While the long-term efficacy of the agents delivered and studied to date remains challenging to evaluate, CED is a promising technique for treating intracranial tumors [77,78,79]. CED bypasses the challenge posed by the BBB. CED does not rely on a steep concentration gradient to drive flow allowing the delivery of a homogenous drug concentration throughout its volume of distribution. Importantly, CED occurs independently of the agent’s diffusivity or MW [80]. Intrathecal administration of antibodies has been achieved by placing a catheter through the cisterna magna. ^211^At-labeled trastuzumab was intrathecally administered to treat HER2-positive breast carcinoma-derived carcinomatous meningitis in rat models [81]. Referring to nanobodies, VHH-B3a is an inhibitory nanobody of the β-site amyloid precursor protein-cleaving enzyme 1 (BACE1) which has demonstrated homogeneous distribution and in vivo BACE1 inhibition upon intracisternal injection [62]. Another study assessed the greater brain access of a llama-derived A20.1 nanobody—detecting *Clostridium difficile* toxin—than a goat-anti-rabbit IgG by molecular imaging approaches after intrathecal administration [82].

Other physicochemical methods have been explored to increase BBB permeability. Several examples are found in the literature that uses mannitol, a hyperosmotic agent, to improve the permeability of drugs and antibodies through the BBB [83,84]. A nanobody against the actin-binding protein gelsolin (NB11; labeled form ^89^ZrNB(DFO)_2_), a key regulator of actin filament assembly and disassembly, was capable of access into the brain by intra-arterial delivery administration regardless of the BBB status. As a proof of concept, the osmotic BBB opening with mannitol further enhanced BBB permeability and the nanobody brain retention by ~2.5-fold [85]. Furthermore, co-injections of pa2H, a nanobody targeting amyloid-beta peptides, with mannitol allowed in vivo detection of parenchymal and vascular amyloid-beta deposits after compromising BBB integrity [86].

Brain temperature *per se* is a critical factor in controlling homeostasis and permeability of the BBB, as physiologically relevant temperature increments cause permeability between adjacent endothelial cells [87,88]. Naturally occurring hyperthermia-related diseases induced an increased permeability of the BBB and brain vasogenic edema [89]. Hyperthermia-based therapeutical approaches, which have been applied to increase temperatures in the range of 40 to 43 °C in body tissues to improve the clinical benefits of other treatment modalities, could be exploited to induce a transient disruption of the BBB [90]. Hyperthermia derived from the magnetic excitation of nanoparticles has demonstrated relevant but reversible opening of the BBB in the absence of inflammation processes [91,92]. This heat-driven transient BBB disruption has allowed higher delivery of bioactive therapeutic compounds into the brain parenchyma [93]. However, to our knowledge, nanoparticles have been used as shuttles of nanobodies through the BBB, instead of facilitating their passage upon magnetic hyperthermia in a two-step (first nanoparticles, later nanobodies) administration approach [66,94].

### 4.3. Receptor-Mediated Transcytosis

BMECs form adherent and tight junctions at the neurovascular unit that impose transcytosis as the only way to transport molecules from the bloodstream through their capillary wall into the CNS or vice versa, except for those small and lipophilic. Molecules crossing through the BBB by receptor-mediated transcytosis need to comply several criteria to satisfy CNS diagnostic and/or therapeutic purposes: (i) their molecular target at the BBB may be a transmembrane receptor whose expression could depend on pathophysiological events; and (ii) neither the normal function of their receptor at the BBB, nor its interaction with natural ligands, may be compromised while binding of these molecules. There is a wide range of nanobodies targeting cell plasma membrane proteins [95]. Then, receptor-mediated transcytosis can be exploited to favor BBB penetration of nanobodies [11] (Figure 3).

FC5 (GenBank no. AF441486) and FC44 (GenBank no. AF441487), the first nanobodies reported to cross the BBB by receptor-mediated transcytosis, were isolated by panning of a non-immune llama phage-displayed nanobody library on BMECs [96]. FC5 was shown to bind the alpha(2,3)-sialoglycoprotein receptor of the luminal human corneal endothelial cells, which initiates the formation of clathrin-coated vesicles for actin- and phosphatidylinositol 3-kinase (PI3K) dependent transcytosis [97]. FC44 was shown to bind ~36 kDa proteins of the BMECs [96]. FC5 and FC44 transmigrate across the BBB in vivo, reaching a 20- to 40-fold increase of the CSF/plasma ratio compared to two control nanobodies (EG2 and A20.1) [98]. Nanobodies crossing the BBB can be exploited as molecular BBB shuttles to deliver diagnostic or therapeutic cargos into the brain. Monovalent and bivalent fusions of FC5 to the human Fc domain increase transport rates across in vitro BBB models. Furthermore, their conjugation to leu-encephalin analog dalargin (mono-FC5-hFc-Dal and bi-FC5-hFc-Dal) enhanced the brain delivery of this BBB-impermeable neuropeptide, enabling pharmacological response [55].

The insulin-like growth factors (IGFs) and their receptors in CNS development and function have been extensively studied [99]. The insulin-like growth factors may affect brain function by either local tissue expression or peripheral circulating peptides crossing the BBB. The BBB uptake of circulating insulin-like growth factors involves the IGF1R and the low-density lipoprotein receptor-related protein 1 (LRP1). Thus, IGFs can reach the CSF and specific regions of the brain. IGF1R has also emerged as a target to promote receptor-mediated transcytosis [100]. VHH-IR5, a nanobody targeting the type 1 insulin-like growth factor receptor 5 (IGF1R), does not appear to interfere with IGF1R kinase activity and its interactions with the endogenous ligand IGF-1. VHH-IR5 targets with high affinity a linear epitope at the α-helix in the C-terminal segment (α-CT) of IGF1R, sharing the binding site of IGF-1 in an independent and non-competitive manner [100,101], suggesting a potential role of the α-CT in VHH-IR5 transcytosis across the BBB.

Eight nanobodies with differential recognition and affinities to vascular and parenchymal amyloid-beta deposits have been described [102]. Among them, nanobodies termed ni3A and ni8B, both isolated from a non-immune phage display library, recognize Aβ1-42 peptide and showed specific detection of vascular but not parenchymal amyloid-beta deposits [86]. ni3A and ni8B are actively transported through the BBB in vitro and with higher transmigration velocity than FC5. The N-terminus of ni3A contains a unique combination of three amino acids [R15-D-G-D] that enormously facilitates its ability to cross the BBB in vitro. Still, the specific receptor is as yet unknown [103]. pa2H is a nanobody obtained from a llama phage display library obtained after immunization with post-mortem brain parenchyma of a patient with Down’s syndrome. pa2H has affinity preference for Aβ1-42 and Aβ1-40 peptides and showed specific detection of vascular and parenchymal amyloid deposits. Nanobody pa2H showed slight cerebral uptake in vivo due to its low blood circulation time, and further modifications to promote its BBB penetration are described below [86].

VCAMelid was engineered for bivalent/bispecific binding (BiVCAMelid) or bispecific binding by fusing an albumin-binding nanobody arm (VCAM/ALB8) resulting in enhanced affinity or increased blood circulation half-life, respectively. Both characteristics enabled better brain uptake in comparison with the monovalent format. Based on brain uptake levels, VCAMelid may perform receptor-mediated transcytosis to cross the BBB, as BMECs have a high basal expression of VCAM-1 [59].

Other nanobodies have been shown to exploit receptor-mediated transcytosis to reach the CNS. Efforts to target prion protein (PrP) with camelid antibodies led to nanobody PrioV3, which showed efficient transmigration across the BBB and diffusion in brain parenchyma when systemically administered in mice by binding its specific antigen by clathrin-mediated endocytosis [104]. Furthermore, PrioV3 enters the cell membrane and targets PrP^C^ and PrP^Sc^, abrogating their accumulation [104,105]. Another nanobody against human and murine PrPs, Nb484, inhibited prion propagation in vitro, but its potential to cross the BBB has not yet been elucidated [44].

To our knowledge, TXB2 is the only type II VNAR described to cross the BBB by receptor-mediated transcytosis [106]. It was selected from a semisynthetic shark phage-displayed VNAR library by in vitro panning on the recombinant human transferrin receptor (TfR1) ectodomain (rh-TfR1-ECD), followed by in vivo selection in mice. The TfR1 is responsible for the transport of iron through the BBB into the brain parenchyma. It plays an essential role in maintaining iron homeostasis for proper brain function [107].

Targeting transferrin receptors at the BBB improves the uptake of immunoliposomes and subsequent cargo transport into the brain parenchyma [108]. Fusion of bivalent TXB2 to human Fc domain (TXB2-hFc) targets TfR1 with high affinity and cross-species reactivity. TXB2-hFc achieved 13- to 21-fold increased brain levels compared to controls and had no effect on TfR1 expression or adverse responses elicited by other high-affinity, bivalent TfR1 antibodies. The further conjugation of neurotensin (TXB2-hFc-NT) showed a specific exerted hypothermic response. Then, the TXB2 can be exploited to cross the BBB by receptor-mediated transcytosis. For example, the fusion of TXB2 to the amyloid-β antibody bapineuzumab (Bapi-TXB2) improved brain uptake of this therapeutic antibody in a threefold manner [56]. Furthermore, a nanobody targeting mouse transferrin receptor (mTfR) has been fused to neurotensin, demonstrating BBB-transport by receptor-mediated transcytosis, as subsequent hypothermic response was elicited after intravenous injection [109].

### 4.4. Adsorptive-Mediated Transcytosis

Adsorptive-mediated transcytosis is triggered by electrostatic interaction between cationic molecules and anionic microdomains on the cytoplasmic side of the BMECs membrane, providing an alternative route for brain delivery of drugs. The BBB dispenses both the potential for binding and uptake of cationic molecules to the luminal surface of BMECs and then for exocytosis at the abluminal surface. Adsorptive-mediated transcytosis of nanobodies requires them to have a basic isoelectric point (pI) for their delivery into the brain parenchyma [110,111,112].

Several nanobodies with a high pI (~9.5) have been described to spontaneously penetrate the BBB [13,60] (Figure 4). Basic nanobody targeting GFAP (mVHH E9) with pI = 9.4, and also its fusion with enhanced green fluorescent protein reporter (mVHH E9-GS-EGFP, pI = 9.3), have been shown to spontaneously cross the BBB and specifically label this intracellular target in brain astrocytes in vivo [60]. Anti-Aβ40/anti-Aβ42 R3VQ (pI > 8.3) and anti-pTau A2 (pI > 9.5), two nanobodies with basic pI isolated from a phage display library, have demonstrated their ability to cross the BBB and recognize the extracellular amyloid deposits and intracellular tau neurofibrillary tangles in mouse models of Alzheimer’s disease. R3VE, a nanobody variant with neutral pI, presented limited brain penetration [61]. R3VQ has been used as a scaffold for the modeling of magnetic resonance imaging probes (R3VQ-S-(DOTA/Gd)_3_), conserving its BBB permeability and targeting capacities [113].

### 4.5. Shuttle-Mediated Transcytosis

Multiple carriers can be used as shuttles to promote the transport of molecules, including nanobodies, through the BBB. These include cell-penetrating or trojan peptides, liposomes, exosomes, and nanoparticles, among others (Figure 5). In shuttle-mediated transcytosis the nanobody, is passively transported taking advantage of a shuttle that is able to cross the BBB by different routes.

#### 4.5.1. Cell-Penetrating Peptides (CPPs)

CPPs are typically short sequences of cationic amino acids with efficient translocation capacity across cell membranes, which enables cellular uptake and BBB-transport of a variety of molecular cargos (small molecules, peptides, proteins, antibodies, siRNA, DNA, plasmids, and nanoparticles), even though their non-selective penetration [114]. The use of CPP penetratin as a connector between the light- and heavy-chain variable domains of a single-chain antibody fragment (scFv V5B2), which targeted the pathological form of the prion protein (PrP^Sc^), allowed scFv to cross through the BBB and specific targeting of brain cells in mice [115]. Further, penetratin was fused to the N-terminus of two scFvs against α-synuclein oligomers (scFv D5 and scFv 10H), demonstrating transport across the BBB following systemic delivery [116]. Although fusions of CPPs to scFvs have proved their potential to cross through the BBB, the permeability of fusions of CPPs to nanobodies remains to be elucidated. The examples in the literature are limited only to the study of access to the cytoplasm. Ligation of cyclic arginine-rich CPPs (cTAT and cR10) to nanobodies (GBP1 and GBP4) targeting GFP-modified proteins were capable not only of binding and relocalizing intracellular antigens, but also to load large recombinant proteins into living cells. These CPP-conjugated nanobodies directly cross the cell plasma membrane and displace their targets to the nucleolus with long-term stability [117]. Furthermore, a nanobody directed against EGFR (VHH 7D12) was conjugated at the C-terminus to the CPP hLF, derived from human lactoferrin. This construct enhanced VHH 7D12 transfer through the cell membrane upon EGFR internalization in a clathrin-mediated endocytosis process [118,119]. CPP-conjugated nanobodies in addition to entering from the luminal side to the cytoplasm of BMECs require to exit through the abluminal side of the endothelium, ensuring BBB penetrance (Figure 5a).

Other peptides have been used for the delivery of molecules into the brain parenchyma. Fusion of a model therapeutic enzyme (e.g., α-L-iduronidase) with a receptor-binding peptide from ApoE has demonstrated its potential to circumvent the BBB by binding to LRP1 and treat neurological disorders in vivo upon transcytosis [120]. The inclusion of peptide analog of ApoΕ3 in anti-transferrin mAb decorated liposomes enhanced their potential to cross the BBB in vitro [121,122]. ApoE-derived peptides could be employed for transendothelial BBB delivery of nanobodies.

#### 4.5.2. Liposomes

Decorated liposomes have provided a suitable system for the specific delivery of nanobodies through the BBB, increasing blood residence and bioavailability. Efforts to elongate the blood half-life of the amyloid-beta peptide targeting nanobody pa2H [86] have been made. Fusion of nanobody pa2H to the Fc portion of the human IgG1 antibody (bivalent VHH-pa2H-Fc-DTPA-111In) prolonged blood circulation time but did not improve brain uptake [123]. However, glutathione targeted PEGylated (GSH-PEG) liposomes encapsulating this nanobody labeled with DTPA-111In (GSH-PEG EYPC VHH-pa2H-DTPA-111In) improved its brain access and retention in amyloid plaques in a mouse model of Alzheimer’s disease [123]. Among others, active transport of GSH through the BBB is carried out by a Na^+^-dependent GSH transporter located at the luminal membrane of brain endothelial cells [124]. A dual-targeting system based on transferrin receptor (TfR)-binding peptide T12 and anti-PD-L1 nanobody modified liposomes (T12/P-Lipo) has been demonstrated to efficiently co-deliver simvastatin and gefitinib on brain metastasis of non-small cell lung cancer in mice, repolarizing and sensitizing the tumor-associated macrophages towards chemotherapy [125] (Figure 5b).

#### 4.5.3. Nanoparticles

Nanoparticles are small molecules with sizes ranging from 1–1000 nm. Nanoparticles are considered one of the most promising and versatile drug delivery systems. They can protect therapeutic agents while efficiently delivering them into the damaged areas [126,127]. Several nanoparticles formulations have been administered intravenously in healthy animals proving their efficacy in crossing the BBB, mainly when modified with ligands or surfactants [126,127]. Advancements in the BBB-penetrating nanoplatforms for brain related disease diagnostics allows their modification to exploit adsorptive-mediated transcytosis to migrate through the BBB [126,127] (Figure 5c).

Single-walled carbon nanotubes (CNTs) have been decorated with nanobodies binding GFP, combining their photophysical properties and targeting capabilities, respectively. Despite their different orientation, randomly conjugated [128] or site-specific labeled [129] single-walled CNT-nanobodies could detect GFP in vitro and in vivo. Brain delivery of CNTs across the BBB has been demonstrated using in vitro BBB models and after systemic administration in mice. Apart from their unique internalization ability by directly crossing biological membranes, most CNTs underwent endocytosis (macropinocytosis)-mediated BBB transcytosis, although the absence of targeting moieties in their structure [130]. Nanoparticle-nanobodies have been used as a targeted delivery system for brain disease theranostics. Multi-walled CNTs, labeled with nanobodies targeting the pearl gentian grouper nervous necrosis virus (PGNNV) on the outermost layer (MWCNTs-PEI-R-Nb), probed their ability to cross the BBB and treat virus-induced CNS disease in zebrafish larvae models after systemic exposure [94]. Systemic injection of anti-IGFBP7 nanobody conjugated to Fe_3_O_4_ PEGylated Cy5.5-labeled nanoparticles led to higher rates of average fluorescence concentration in the orthotopic implanted GBM region in vivo, in comparison with Cy5.5-labeled anti-IGFBP7 nanobody [66]. Owing to their multivalent nature, the increased binding avidity for targets that present nanoparticle-nanobodies conjugates may further enhance their BBB penetrance [131,132]. Other encapsulating methods have been used to deliver antibodies to the CNS using a thin shell of polymer containing choline and acetylcholine receptor analogs [133].

### 4.6. Somatic Gene Transfer of Nanobodies into the Brain Parenchyma

Novel gene transfer strategies for delivering antibodies directly into de CNS have been developed, allowing long-term local production of therapeutic concentrations [134]. VHH-B9, a nanobody targeting BACE1, has demonstrated long-term therapeutic effects by inhibiting amyloid-beta peptides production after adeno-associated virus (AAV)-based delivery in a mouse model of Alzheimer’s disease [135]. However, AAV-VHH-B9 vector was administered directly into the hippocampus.

Genetically engineering the viral capsid of AAV vectors is commonly used to improve their transduction or achieve tissue tropism [136]. Several groups have developed recombinant AAV vectors with enhanced somatic gene transfer to the CNS after intravenous delivery [137,138]. These novel vectors include BBB shuttle peptides that enhance AAV transduction in the brain after systemic administration. Therapeutic systemic injection for vector-based delivery of nanobodies will be possible by using these new AAV vectors, which traverse the BBB with high efficacy and enable widespread CNS gene transduction [138].

## 5. Discussion and Future Perspectives

The BBB is a highly selective and regulated filter that controls and limits the passage of molecules from the bloodstream into the brain parenchyma and vice versa. Many drugs directed against CNS targets suffer from a very high rate of failure due to the BBB, limiting the entry of xenobiotics into the brain [139,140]. In contrast to conventional immunoglobulins, which are described as BBB impenetrable when the barrier is intact, some nanobodies can cross through the BBB or be easily modified to favor their penetration.

The integrity of anatomical barriers, including the BBB, is a hallmark of health [12]. While the adult BBB is a selective and restrictive filter, embryonic and newborns BBBs are “immature”. The BBB is a vascular entity *per se*, while angiogenic and vasculogenic processes in fetal and early stages of life provide incomplete or “leaky” blood vessels. Developing cerebral vessels appear to be more fragile than in adults, rendering the developing brain more vulnerable to drugs or toxins [141]. Further studies are required to better determine the penetrance of nanobodies through the BBB in newborns.

Lack of integrity and dysfunction of the BBB occurs in multiple brain pathologies, including cancer, resulting in BBB leakiness and increased xenobiotics uptake. This fact explains, at least in part, the clinical benefit shown by some nanobodies directed against CNS targets [65] (ClinicalTrials.gov NCT03331601). Furthermore, multiple neurological diseases are caused by various inherited monogenic genetic mutations affecting individual cell types within the BBB. These genetic defects cause specific alterations in the development and maintenance of the BBB homeostasis, resulting in BBB disruption [17], which could favor the entrance of nanobodies into the CNS.

As for other drugs, especially when treating brain tumors, nanobodies can be directly placed into the brain parenchyma by CED. However, the long-term evaluation of the efficacy of delivered molecules remains challenging and homogenous concentration throughout all their distribution cannot be reached [77,78,79]. Several methods can be used to increase brain uptake of nanobodies by eliciting a direct and transient BBB impairment to bypass its restrictions. These include osmotic BBB opening with chemical agents such as mannitol [83,89,90]. Other agents merit further investigation to evaluate if they could help nanobodies to cross the BBB. For example, RMP-7 (Cereport) is a bradykinin agonist used to transiently and safely increase BBB permeability through activation of constitutive B_2_ receptors of endothelial cells [142]. Concurrent administration of RMP-7 has demonstrated its potential to improve the efficacy of chemotherapeutic drugs in brain tumor patients [143,144]. Regadenoson (Lexiscan), an adenosine A_2A_ receptor agonist, facilitated the entry of therapeutic compounds by temporally modulating BBB permeability [145,146]. Like mannitol, RMP-7 and regadenoson could be systemically co-administered to improve the access of nanobodies into the brain parenchyma.

Other approaches to disrupt the BBB have been explored. These include, among others, transcranial ultrasound pulses using micro-/nano-bubbles [147]. The development of nanobody-coupled microbubbles and nanobubbles conjugates as novel molecular ultrasound contrast agents has been reported for other non-CNS related diseases. Nanobubbles coupled with nanobodies targeting the prostate-specific membrane antigen (PSMA) facilitated prostate cancer imaging by ultrasonography [148]. Anti-G250 nanobody-bearing targeted nanobubbles improved ultrasound imaging of renal cell carcinomas in mouse models [149,150]. Anti-eGFP (cAbGFP4) and the clinically translatable anti-VCAM-1 (cAbVCAM1-5) nanobodies have been employed for tailoring microbubbles to improve their targeting and imaging potential both in vitro and in vivo [151,152]. This approach merits further studies. Ultrasounds drive the oscillation of microbubbles upon systemic administration and the consequent mechanical stress on the endothelial cells of the BBB may cause permeability leakage by acoustic cavitation [153]. Microbubbles can be destroyed upon exposure to transcranial ultrasound pulses resulting in acoustic forces inducing vessel permeability and BBB permeation for a 6–8 h time range [147]. Ultrasound-mediated BBB opening has been shown to enhance the brain delivery of therapeutically relevant formats of a tau protein-specific antibody in preclinical models [154]. In this direction, microbubbles and nanobubbles could be loaded with nanobodies within their shell to enhance nanobody passage through the BBB.

BBB disturbance following localized hyperthermia in rats was described in the 90s [155]. More recently, several reports indicated a reversible BBB opening by hyperthermia using gold nanoparticles, combined with near-infrared light, or magnetic nanoparticles, which can produce local hyperthermia when applying an external alternating magnetic field [156].

BMECs form adherent and tight junctions that impose transcytosis as the unique way to transport molecules from the bloodstream through their capillary wall into the brain parenchyma or vice versa, except for some lipophilic and smaller than 400 Da [28]. The latest advances in the biology field contributed to the development of a toolbox of molecular strategies which benefit from BBB physiology to allow brain uptake of nanobodies.

Molecules crossing through the BBB by receptor-mediated transcytosis need to target a transmembrane receptor expressed at the BBB cells and be innocuous for the receptor interaction with natural ligands. Receptor-mediated transcytosis can be utilized as a pathway for BBB permeability of nanobodies. As described above, the FC5 nanobody targets the alpha(2,3)-sialoglycoprotein receptor initiating the formation of clathrin-coated vesicles for actin and PI3K dependent transcytosis [96]. Of importance, nanobodies crossing the BBB can be exploited as molecular BBB shuttles to deliver other cargos, including other nanobodies. For example, FC5 conjugation to leu-encephalin analog dalargin enables the brain to deliver this BBB-impermeable neuropeptide, rendering pharmacological response [55].

Several nanobodies with a high pI (~9.5) have been described to spontaneously cross the BBB by adsorptive-mediated transcytosis [13,60,61,125]. CPPs and trojan peptides crossing the BBB can also be fused to nanobodies to favor entrance into the CNS by using this route [157]. Of particular relevance, adsorptive-mediated transcytosis of nanoparticles [66,94] and decorated or functionalized liposomes [54] can be employed to deliver nanobodies into the brain. In this direction, extracellular vesicles and exosomes are being intensively studied to be used as carriers by exploiting adsorptive-mediated transcytosis for the non-invasive delivery of molecules across the BBB [158]. Advantageously, they exhibit the intrinsic capacity of transferring a great variety of molecules with minimal immunogenicity.

Other strategies to circumvent BBB merit attention. Somatic gene transfer of nanobodies coding sequences using viral vectors [134] could help achieve long-term local production of therapeutic concentrations in the CNS. Other CPP-functionalized liposomes and nanoparticles for somatic gene transfer [159] could represent another route of somatic gene transfer of nanobodies coding sequences into the CNS.

Extracellular vesicles are small membrane vesicles implicated in local and systemic cell–cell communication through the horizontal transfer of information (i.e., mRNAs, microRNAs, and proteins) [160,161,162]. Exosomes (30–100 nm) derive from the luminal membranes of multivesicular bodies and are constitutively released via fusion with the cell membrane. In contrast to liposomes, exosomes exhibit the intrinsic capacity of transferring molecules of almost any chemical nature with low immunogenicity. Exosomes are being extensively explored as a means of drug discovery and delivery and serve as shuttles for the non-invasive delivery of molecules across the BBB to the CNS [158].

Modified exosomes have been utilized as a system for the delivery of proteins and RNA to the brain (Figure 5b). Firstly, successful protein delivery through the BBB was achieved from macrophage-derived exosomes loaded with catalase [163] or with brain-derived neurotrophic factor (BDNF) decorating their surface [164]. These naïve exosomes are free of brain homing peptides and mediate BBB penetrance through the interaction with lymphocyte function-associated antigen (LFA)-1, intercellular adhesion molecule (ICAM)-1, and C-type lectin receptors. Exosomes have also been functionalized with targeting peptides postproduction, including a multifunctional peptide (L-4F) that enables to anchor itself or other peptides expressed on the BBB and glioma cells (e.g., LDL peptide, targeting the low-density lipoprotein receptor) to the exosome membrane [165]. This strategy directed systemically injected exosomes into the brain and allowed a successful targeting of methotrexate-loaded exosomes to glioma cells in mouse models of glioma. Secondly, siRNA and microRNA brain uptake and target silencing were reached by exosomes derived from dendritic cells [166,167], mesenchymal stem cells [168], or human HEK293T cells [169] which harbored different siRNA and microRNA inside. However, these exosomes needed to incorporate the 29-mer Rabies Virus Glycoprotein (RVG) peptide or the transferrin receptor-binding peptide T7 for specific binding to brain cells.

Altogether, exosomes hold many potential advantageous features compared to other synthetic delivery systems such as liposomes and nanoparticles regarding their intrinsic properties, biodistribution, and ability to deliver a functional cargo into targeted cells [170]. However, an important limitation of the exosomes resides in their rapid clearance and low accumulation in target tissues and cells [171]. Therefore, exosomes have been modified to improve their delivery towards their target sites of action. Display of glycosylphosphatidylinositol-anchored anti-EGFR nanobodies on extracellular vesicles has been produced to promote tumor cell targeting [172]. In the same direction, the external adhesion or the encapsulation of nanobodies (e.g., protein or RNA-encoding nanobody molecules for gene transfer) offers new alternative approaches for the nanobody delivery through the BBB to the brain.

The actual emerging field of mRNA vaccines could pave the way for a nanobody-coding vaccine with clinical applications in the future. In addition, an innovative strategy to deliver molecules through the BBB could be the production and release of these molecules from chimeric antigen receptor T (CAR-T) cells, which cross the BBB. In the immunotherapy field, nanobodies have been extensively applied as the antigen-binding part of CAR-T cells [173]. However, some groups have shown the efficacy of secreting nanobodies by CAR-T cells to circumvent their restrictive access in solid tumors [174]. The use of CAR-T cells as a vehicle for local delivery of nanobodies in the CNS could avoid obstacles imposed by the BBB, reaching the brain parenchyma.

Alternative physiological routes used to treat CNS pathologies should be considered. The blood–cerebrospinal fluid (CSF) barrier (BCSFB) is consists of choroid plexus epithelial cells and the arachnoid membrane. In addition to the BBB, the BCSFB constitutes a second dynamic transport interface, displaying similar transcytosis mechanisms, responsible for the regulated passage of molecules to the CNS [175,176]. The BCSFB could be a plausible route for the transport of nanobodies into the brain [6]. Otherwise, incipient knowledge describes the presence and role of lymphatic vasculature in the CNS [177]. Conventionally, the lymphatic system connects the circulatory and immune systems, acting in unison with blood vessels to exchange molecules and immune cells within tissues. The brain immune privilege was commonly believed that was due to the lack of a lymphatic drainage system. However, detection of lymphocyte and tumor cell trafficking from the brain to the cervical lymph nodes suggested that there could be a direct route of passage into the lymphatic system in mice [178,179]. A lymphatic system along with the draining cerebral sinuses in mice [180,181] and humans [182] has been recently discovered sifting paradigms. These lymphatic vessels serve as conduits to the cervical lymph nodes to exchange fluid and immune cells with the CSF. Further studies are required to determine if these findings could provide a background for attempts to use nanobodies directed against CNS targets via the CNS lymphatic route.

## 6. Conclusions

We live in a new era for nanobodies. ALX-0681 (Caplacizumab), a bivalent nanobody [52] for the treatment of thrombotic thrombocytopenic purpura patients, recently received approval from the US Food and Drug Administration (FDA) and the European Medicines Agency (EMA), giving a boost to domain antibodies in research and clinics. Trespassing the BBB and targeting the brain parenchyma is still a major goal for nanobodies to treat, diagnose and monitor neurological disorders and CNS pathologies. While the BBB represents the bottleneck for CNS drug development, imposing a limitation to deliver targeted therapies to the brain parenchyma, several strategies—as shown here—can be exploited to fulfill the therapeutic, diagnostic, and theragnostic promise of single-domain antibodies.

## Figures and Tables

**Figure 1 biomolecules-11-01131-f001:**
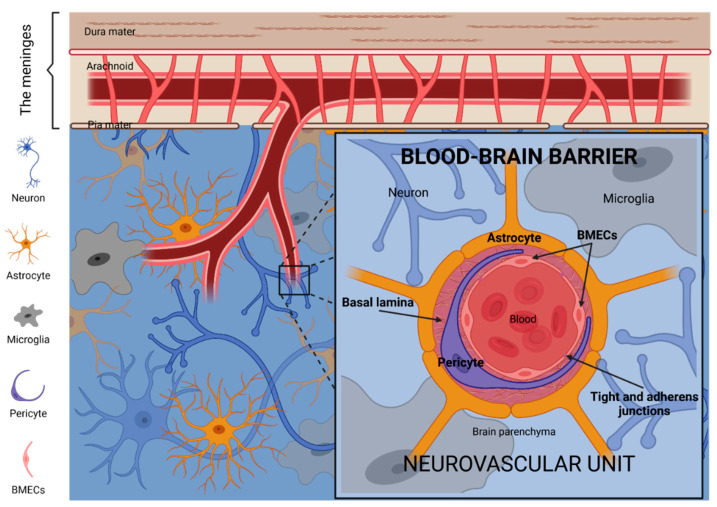
The blood–brain barrier (BBB). The neurovascular unit is the principal anatomical and functional structure of the BBB. BMECs, thoroughly closed between them by tight and adherens junctions, and surrounded by large pericytes and the basal lamina matrix, constitute the vessels of the neurovascular unit. Neurons, microglia and the endfeet of astrocytes constitute the microenvironment which supports and communicates with the vessel components of the neurovascular unit. Image created with BioRender.com (accessed on 17 June 2021).

**Figure 2 biomolecules-11-01131-f002:**
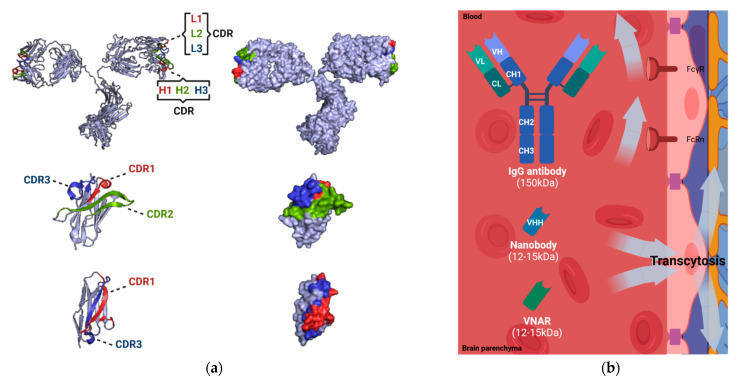
Comparison of IgG antibodies and single-domain antibodies (nanobodies and VNARs). (**a**) Molecular 3D structures of canonical IgG antibody (PDB reference: 1IGT), nanobody (PDB reference: 1I3V), and VNAR (PDB reference: 4HGK). The complementarity determining regions (CDRs), known to be responsible for antigen recognition, are displayed (CDR1, in red; CDR2, in green; CDR3, in blue) [37,38]. (**b**) Delivery of IgG antibodies and single-domain antibodies (nanobodies and VNARs) at the BBB. The permeability of IgG antibodies is limited due to the presence of Fc receptors (Fcγ receptor, FcγR; neonatal Fc receptor, FcRn). The permeability of single-domain antibodies (nanobodies and VNARs) is facilitated by physiological transcytosis mechanisms. Image created with PyMOL2 and BioRender.com (accessed on 17 June 2021).

**Figure 3 biomolecules-11-01131-f003:**
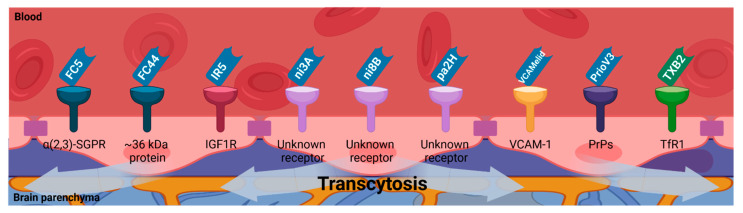
Single-domain antibodies (nanobodies and VNARs) BBB permeability due to receptor-mediated transcytosis. Several membrane receptors which are present on the surface of the endothelial cells of the BBB have been proposed to mediate the transcytosis of single-domain antibodies (nanobodies and VNARs) to transport them into the brain parenchyma (alpha(2,3)-sialoglycoprotein receptor (α(2,3)-SGPR); insulin-like growth factor 1 receptor (IGF1R); vascular cell adhesion molecule 1 (VCAM-1); prion proteins (PrPs); transferrin receptor-1 (TfR1)). Image created with BioRender.com (accessed on 17 June 2021).

**Figure 4 biomolecules-11-01131-f004:**
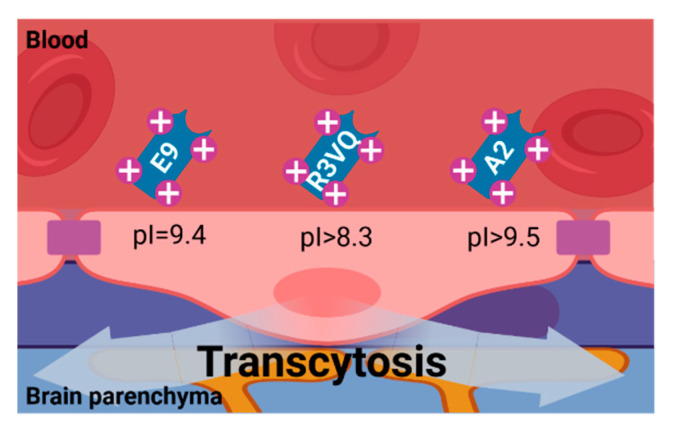
Nanobody BBB permeability due to Adsorptive-Mediated Transcytosis. The positive charged surface of single-domain antibodies (nanobodies and VNARs) with basic pI (~9.5) allow them to spontaneously interact with the cell membrane of the endothelial cells of the BBB, mediating their transcytosis into the brain parenchyma. Image created with BioRender.com (accessed on 17 June 2021).

**Figure 5 biomolecules-11-01131-f005:**
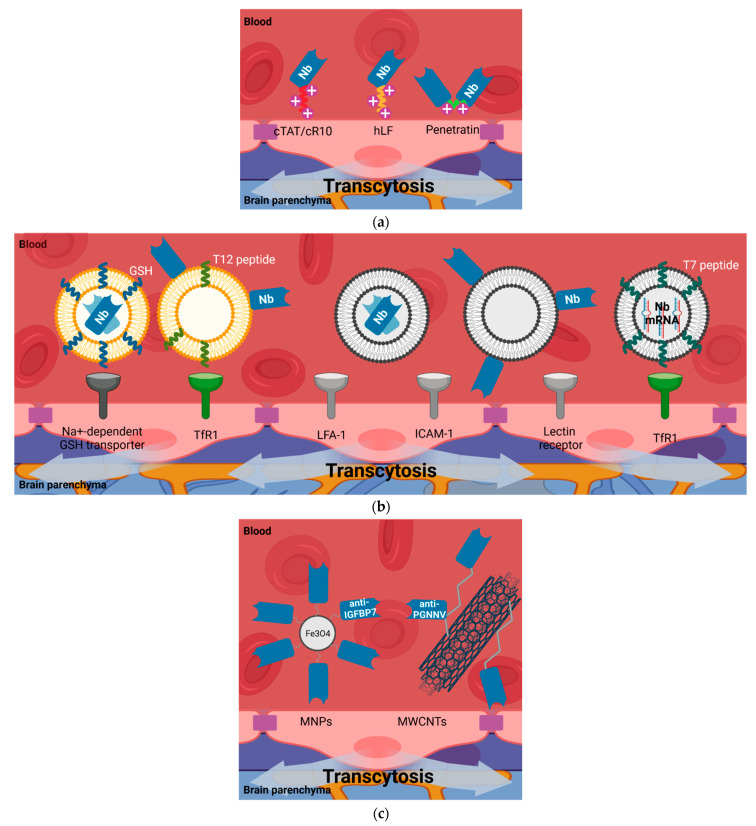
Nanobody BBB permeability due to shuttle-mediated transcytosis. Several molecular entities have been utilized to facilitate the delivery of single-domain antibodies (nanobodies and VNARs) through the BBB into the brain parenchyma. (**a**) Cell-penetrating peptides (CPPs) carrying nanobodies. The positive charges on their surface mediates their interaction with the cell membrane of the endothelial cells of the BBB. (**b**) Liposomes and extracellular vesicles carrying nanobodies. The lipophilic nature of these vesicles, in combination with the presence of peptides targeting BBB receptors, allow the brain uptake of nanobodies decorating their inside part or their surface (transferrin receptor-1 (TfR1); lymphocyte function-associated antigen-1 (LFA-1); intercellular adhesion molecule-1 (ICAM-1)). (**c**) Nanoparticles carrying nanobodies. The multivalent nature of these hybrids allow better BBB penetrance due to an increased binding avidity for targets and blood half-life (magnetic nanoparticles, MNPs; multi-walled carbon nanotubes, MWCNTs). Image created with BioRender.com (accessed on 17 June 2021).

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
