# Peer review of "Transportation of Single-Domain Antibodies through the Blood–Brain Barrier"

_biomolecules, 2021, doi:10.3390/biom11081131_

Round 1

Reviewer 1 Report

This review paper entitled "Transportation of Nanobodies through the Blood-Brain Barrier" is interesting and well organized. The manuscript also provides an up-to-date review about possible mechanism for drug delivery which can be very useful for several neurodegenerative diseases, overcoming drug-resistance and certain brain tumors. 

Minor comments:

  1. Abstract: Camelidae add a few words to define it
  2. Adding a brief introduction before section 1 is needed, give an overview about what is known so far, rational for this review and so on
  3. I think discussing several disease such as Alzheimer's disease and brain tumors in more details rather than mixing all the diseases in few sentences might help the reader appreciate the information easier, you might also create a summary table and discuss few disease and the application of nanobodies, you might also discuss the impact of disease on possible receptor expression and/or function
  4. Several acronyms are used needs to be spelled out in their first appearance (eg. line 57 anti-PD-1m anti-CTLA-4), line 336 PrP and so on.
  5. Amyloid beta 42 is spelled differently lines 186-188
  6. Overall, this is a nice overview and well-written review 

Reviewer 2 Report

see attached file

Reviewer 3 Report

see attached file

Round 2

Reviewer 2 Report

The quality and readability of the review greatly improved. Many thanks for the revisions.

Please check the following remaining issues:

  • The spelling/grammar of the revisions is sometimes poor. please check carefully all adaptations, e.g.:
    • Line 131: "CDR3 being the least the main contributor "
    • Line 140: " their limited immunogenicity feasibility "
    • Line 435: " is passively transported that a shuttle that is able to cross "
    • Line 599: 'naobodies'
    • also check the rest of the text please.
  • Although the rebuttal claims that "permeability" in line 191 is replaced by "integrity", it isn't.
  • " However, if this is not sufficient, we suggest replacing “carrier”-mediated transcytosis with “shuttle”- mediated transcytosis".? Please do so, it will be less confusing. 

thanks
